# Portland and Belite Cement Hydration Acceleration by C-S-H Seeds with Variable *w*/*c* Ratios

**DOI:** 10.3390/ma15103553

**Published:** 2022-05-16

**Authors:** Alejandro Morales-Cantero, Ana Cuesta, Angeles G. De la Torre, Oliver Mazanec, Pere Borralleras, Kai S. Weldert, Daniela Gastaldi, Fulvio Canonico, Miguel A. G. Aranda

**Affiliations:** 1Departamento de Química Inorgánica, Cristalografía y Mineralogía, Universidad de Málaga, 29071 Málaga, Spain; alejandrom@uma.es (A.M.-C.); a_cuesta@uma.es (A.C.); mgd@uma.es (A.G.D.l.T.); 2Master Builders Solutions Deutschland GmbH, Albert-Frank Str. 32, 83308 Trostberg, Germany; oliver.mazanec@mbcc-group.com (O.M.); kai-steffen.weldert@mbcc-group.com (K.S.W.); 3Master Builders Solutions España S.L.U., Carretera de l’Hospitalet, 147-149, Edificio Viena, 1ª Planta, 08940 Cornellà de Llobregat, Spain; pere.borralleras@mbcc-group.com; 4Research & Development, Buzzi Unicem, Via Luigi Buzzi 6, 15033 Casale Monferrato, Italy; dgastaldi@buzziunicem.it (D.G.); fcanonico@buzziunicem.it (F.C.)

**Keywords:** CO_2_ footprint, accelerators, C-S-H nano-seeding, ettringite, Rietveld analysis

## Abstract

The acceleration of very early age cement hydration by C-S-H seeding is getting attention from scholars and field applications because the enhanced early age features do not compromise later age performances. This acceleration could be beneficial for several low-CO_2_ cements as a general drawback is usually the low very early age mechanical strengths. However, the mechanistic understanding of this acceleration in commercial cements is not complete. Reported here is a contribution to this understanding from the study of the effects of C-S-H gel seeding in one Portland cement and two belite cements at two widely studied water–cement ratios, 0.50 and 0.40. Two commercially available C-S-H nano-seed-based admixtures, i.e., Master X-Seed 130 and Master X-Seed STE-53, were investigated. A multi-technique approach was adopted by employing calorimetry, thermal analysis, powder diffraction (data analysed by the Rietveld method), mercury intrusion porosimetry, and mechanical strength determination. For instance, the compressive strength at 1 day for the PC (*w*/*c* = 0.50) sample increased from 15 MPa for the unseeded mortar to 24 and 22 MPs for the mortars seeded with the XS130 and STE53, respectively. The evolution of the amorphous contents was determined by adding an internal standard before recording the powder patterns. In summary, alite and belite phase hydrations, from the crystalline phase content evolutions, are not significantly accelerated by C-S-H seedings at the studied ages of 1 and 28 d for these cements. Conversely, the hydration rates of tetracalcium alumino-ferrate and tricalcium aluminate were significantly enhanced. It is noted that the degrees of reaction of C_4_AF for the PC paste (*w*/*c* = 0.40) were 10, 30, and 40% at 1, 7, and 28 days. After C-S-H seeding, the values increased to 20, 45, and 60%, respectively. This resulted in larger ettringite contents at very early ages but not at 28 days. At 28 days of hydration, larger amounts of carbonate-containing AFm-type phases were determined. Finally, and importantly, the admixtures yielded larger amounts of amorphous components in the pastes at later hydration ages. This is justified, in part, by the higher content of amorphous iron siliceous hydrogarnet from the enhanced C_4_AF reactivity.

## 1. Introduction

A major goal in the Portland cement (PC) industry is to attain carbon neutrality in 2050. This ambitious objective requires that several approaches should successfully converge. One of these approaches is to lower the final binder CO_2_ footprint through its mineralogical composition. Here, a leading work [1] identified clinker replacement by supplementary cementitious materials (SCMs) [2,3] as the most promising strategy today and in the near future. To substitute a fraction of the PC clinker by SCMs is nowadays the approach with the lowest performance and economic impact.

However, it is important to have alternative ways to decrease the cement CO_2_ footprint in the medium future while complying with the current codes of practice. These alternative approaches should allow the incorporation of SCMs and ideally yield concretes with longer service lives. One possibility could be belite cements (BCs), which contain alite and belite, like PC, but where the amounts are reversed. The advantages and drawbacks of BCs have recently been reviewed [4]. The key current disadvantage of BCs is their low hydration rate at early ages, leading to poor mechanical performances at three days of hydration or earlier. Therefore, there is a clear need to increase the reactivity of these cements, which is shared by many other low-CO_2_ cements [5,6], such as blends of PCs with SCMs [2,3,7,8]. These binders usually have low mechanical strengths at early hydration ages because of the slow rate of the pozzolanic reaction(s).

There are different approaches to the activation of the hydration of the reactions of PC [9], alkali-activated [10] and lime [11] binders. The acceleration can be carried out by physical (for instance with power ultrasound [12,13]) or by chemical means. Next, the discussion is focussed on the use of acceleration admixtures for PC and BC binders. Some well-known accelerators, such as calcium ionic salts (CaCl_2_, Ca(NO_3_)_2_, Ca(NO_2_)_2_, Ca(SCN)_2_), sodium metasilicate (Na_2_SiO_3_), or sodium aluminate (Na(Al(OH)_4_), significantly promote the early hydration of cements [14,15,16,17,18]. However, they may present durability problems, or they can decrease the mechanical strength performances at 28 days [18,19]. Alkanolamines are well known as grinding aids, and they are now also being used as accelerator admixtures, with quite variable performances depending upon the chemistry and dosage [18,20]. One approach which does not seem to show adverse issues is seeding with water-based stabilized suspensions of C-S-H nanoparticles [15,21,22,23]. It is noted here that the synergistic effects between C-S-H seedings in the presence of triethanolamine, one type of alkanolamines, have very recently been reported [24].

The use of C-S-H seeding for accelerating cement hydration has already been reviewed [18,25,26]. Therefore, only some of the key aspects are discussed next. C-S-H nanoparticle seeding has two main effects in cement hydration [25]. Firstly, these nanoparticles chemically modify the pore solution ion contents (e.g., Ca^2+^, Al^3+^, SO_4_^2−^, Na^+^, K^+^, etc.) as these ions are partly and effectively adsorbed in the nano-seeds because of their large surfaces. The adsorption of ions from the pore solution, in turn, has two consequences: (i) the changes in the ion concentration in the pore solution creates local concentration gradients which could alter the dissolution pathways in the cementitious systems, and (ii) the C-S-H growth mechanism(s) could be changed because of the adsorbed species. Secondly, C-S-H seeding supplies additional nucleation sites, which are adequate because of the low interfacial energy, which may physically accelerate calcium silicate hydration, also known as the filler effect. C-S-H seeding could be quite different in cements, where there are many ions in the pore solution, when compared to pure alite samples, where the pore solution chemistry is much simpler. Due to this, the present work does not elaborate on the consequences of C-S-H seeding in alite pastes. In this context, the influence of PC mineralogy [27] and cement fineness has been published because, for cements with a larger surface area, the acceleration by C-S-H seeding is comparatively lowered [27]. Several other parameters may also intervene, such as the sulphates [28], the alkaline ion contents, or the hydrating temperature(s) [29]. The positive effects of C-S-H seeding in the cement acceleration of neat PCs have been studied at variable water–cement mass (*w*/*c*) ratios from 0.35 to 0.50 [30].

C-S-H seeding has been employed to accelerate the hydration of a number of low-CO_2_ cements, including BCs [31]. As expected, the role of C-S-H as an admixture is being much more researched in PC–SCM blends: mainly fly ash [24,32,33,34,35,36,37] and, very recently, calcined clays [38,39]. This cement hydration acceleration approach has also been employed for other special purposes such as low temperature curing [40,41].

There are two widely accepted mechanisms for explaining the consequences of C-S-H seeding in cement pastes. Concerning the good mechanical strength (and durability) performances at 28 days or later, synchrotron X-ray diffraction tomography showed that C-S-H addition partly moves away the gel nucleation and growth from the dissolving clinker particles, enhancing secondary nucleation in the capillary porosity of the paste [42]. The shift of the C-S-H gel from the surfaces of alite to the pore space leads to a more homogeneous distribution of C-S-H products in the bulk of the paste [38], which in turn may reduce the porosities, improving the mechanical strength properties and decreasing the permeability of the final binder [43]. Concerning the early age acceleration of cement hydration, it was proposed [21] that alite hydration is accelerated because the synthetic C-S-H seeds provide additional nucleation sites, enabling the C-S-H from alite to rapidly grow.

Within our ongoing investigations, focused on the early age hydration acceleration of low-CO_2_-footprint cements, in a first work [44] the hydration of a PC and two BCs (activated and non-activated) was accelerated by admixtures at a fixed *w*/*c* mass ratio of 0.50. The admixtures included X-Seed 130 (one commercial source of C-S-H nanoparticles) and triisopropanolamine (TIPA). A multi-technique approach was employed, including in situ X-ray synchrotron powder diffraction, which allowed the determination of the average reaction rate of every crystalline phase in the cements. In summary, the hydration rates of the alite and belite phases were not significantly accelerated by C-S-H seeding for the three studied cements. Conversely, the sulphate and aluminate phase dissolutions were very significantly enhanced. This faster dissolution resulted in more rapid ettringite crystallisation, which contributed to the observed improved mechanical strength performances at early ages. Therefore, it was concluded [44] that for the studied cements, the cement hydration acceleration at very early ages by C-S-H seeding was mainly due to enhanced ettringite crystallization.

As the previous results challenged a common understanding about PC hydration acceleration by C-S-H seeding, here the research is enlarged by using the same three cements. In the present work, the effects of two C-S-H seeding admixtures, X-Seed 130 and X-Seed STE53, at two common *w*/*c* mass ratios were investigated. The acceleration of the cement hydration was followed by measuring the mechanical strength performances and the calorimetric curves. After establishing the acceleration signatures, the thermal and Rietveld quantitative phase analyses were carried out to obtain insights about the mechanisms of cement hydration acceleration. Furthermore, mercury intrusion porosimetry was employed to follow the microstructural changes.

## 2. Materials and Methods

### 2.1. Materials

The three cements employed here, (i) a commercial Portland cement, CEM I 42.5 R, PC-42.5, which conforms to EN 197–1; (ii) a Belite cement, CEM I 42.5 N-like, fabricated by Buzzi Unicem SpA (Italy), BC-Buz, which was activated by sulphur during clinkering; and (iii) a non-activated Belite cement, BC-n.a., were already used in a previous publication [44]. Therefore, the cement characterization is given in the Appendix A. Their elemental, mineralogical, and textural analyses are reported in Appendix A, respectively.

For the powder diffraction studies, the hydration-arrested pastes were mixed with a 20 wt% of quartz (SiO_2_, 99.5%, AlfaAesar) as an internal standard in order to determine the amorphous and non-identified components (ACn) [45]. Two commercially available accelerator admixtures, C-S-H gel seed-based, were used. The C-S-H gel nanoparticles were kept dispersed in these admixtures by using the appropriate chemical substances. These materials are Master X-Seed 130 (XS130) and Master X-Seed STE-53 (STE53) from Master Builders Solutions España S.L.U., Barcelona, Spain. STE53 is a recent development aimed at improving both early and late age strengths. Master X-Seed 130 and X-Seed STE-53 have solid contents close to 30%. Moreover, these admixtures also contain alkanolamines.

### 2.2. Pastes and Mortars Fabrications

For the preparation of the pastes, *w*/*c* mass ratios of 0.40 and 0.50 were used. For the isothermal calorimetry study, the pastes were prepared as follows: (i) the water with the admixture was magnetically stirred for 1 min; (ii) the cement was mixed with the diluted suspension; and (iii) the paste was manually shaken for 1 min with a glass bar and then stirred with a vortex mixer for another minute. Finally, the pastes were inserted into the glass ampoules and then into the calorimeter.

The amounts of admixtures added to the pastes, dosed as the received products, were 2 wt% by weight of cement for XS130 and STE53. It is important to note that the amount of water in these admixtures was taken into consideration when carrying out the water dosage, to avoid having slightly larger *w*/*c* ratios in the admixtures containing pastes.

For the powder diffraction study, the pastes were prepared with a stirrer and introduced into the cylinders as previously reported [46]. After 24 h, the specimens were demoulded and kept in water in plastic bottles until the selected hydration age. The cylinders were packed as much as possible, i.e., with the minimum amount of water, to minimize calcium leaching. Finally, the pastes were manually ground, filtrated, and washed with isopropanol twice and finally with diethyl ether.

The mortars were prepared with the following mass proportions: 0.40 or 0.50/1/3 of water/cement/sand, respectively, according to EN 196-1. Again, the water content within the admixtures was taken into account when used. The mortars were cast into moulds of 40 × 40 × 160 mm^3^ dimensions and loaded using a jolting table UTCM-0012, 3R (Montauban, France), as previously reported [46]. The mortars were maintained in the moulds up to 24 h at 20 °C and 99% RH. Finally, the samples were introduced into water up to the selected hydration time in a room at 20 °C.

### 2.3. Analytical Techniques

#### 2.3.1. Mortar Consistency Measurement

The consistencies were determined, for selected samples, by using the shaking table according to the standard EN 1015–3. In brief, the slump device is a truncated metallic cone with a bottom diameter of 100 mm, a top diameter of 70 mm, and a height of 60 mm over a polished surface. The mortars were prepared as indicated above, and the mould was filled and lifted following the standard. The reported spread values are the average of two perpendicular measurements after lifting the cone and carrying out the knocks.

#### 2.3.2. Compressive and Flexural Strengths

The mortars were measured at 1, 7, and 28 days of hydration for the compressive and flexural strengths according to the standard EN 196-1. The three prims were used for the flexural data, and subsequently, the six resulting specimens were used to obtain the compressive data. The reported results are the average of all the measurements. The press was an Autotest 200/10 W model from Ibertest, Madrid, Spain, which worked at a rate of 1.5 MPa·s^−1^.

#### 2.3.3. Isothermal Calorimetry

Heat flow measurements were performed for all the selected pastes up to 7 days of hydration, at 20 °C, using an eight-channel Thermal Activity Monitor (TAM) calorimeter. After introducing the glass ampoules into the calorimeter, 45 min of holding time was needed for thermal stabilization.

#### 2.3.4. Thermal Analysis (TA)

An SDT-Q600 analyser from TA instruments (New Castle, DE, USA) was used for recording the thermal data of the pastes. The data acquisition was performed as follows: (i) from RT to 40 °C; (ii) this temperature was maintained for 30 min; and (iii) from 40 up to 1000 °C using a heating rate of 10 °C/min. The weighed water loss from 40 to 550 °C was considered as bounded water and that from the 550 to 1000 °C range was considered as CO_2_. Finally, the free water calculation was determined using the protocol and equations previously reported [46].

#### 2.3.5. Laboratory X-ray Powder Diffraction (LXRPD)

For the LXRPD data collection of the pastes, a D8 ADVANCE diffractometer from Bruker AXS was used. This equipment is located at SCAI–Universidad de Málaga. The diffractometer has a *θ*/*θ* configuration geometry with Mo-Kα_1_ strictly monochromatic radiation (λ = 0.7093 Å). The samples were prepared by loading the powder between two Kapton foils without pressing. The Rietveld quantitative phase analyses (RQPA) were performed with the GSAS suite of programs and the EXPGUI graphic interface [47].

#### 2.3.6. Calculated Phase and Water Contents Based on Chemical Reactions

The calculations to estimate the amount of free water (FW) and the amount of amorphous C-S-H gel [48] are detailed in the Appendix A. They were made by using the hydration reactions for the different phases (and making some assumptions which are explained in the Appendix A).

#### 2.3.7. Mercury Intrusion Porosimetry (MIP)

For the MIP analysis, cylinders of pastes at 1, 7, and 28 days were prepared, with a 1 cm diameter and a 1.5 cm height. The hydration was stopped by immersing the samples in isopropanol for 3 days. After this time, the pastes were gently dried at 40 °C up to a constant weight (about 5 days). A micromeritics AutoPore IV 9500 porosimeter (Micromeritics Instrument Corporation, Norcross, GA, USA) was used to measure the porosity, using a range from 1 mm down to 4 nm (radius). The applied pressure, in step mode, ranged from 0 to 206 MPa, and a contact angle of 140° [49] was assumed for the calculations.

## 3. Results and Discussion

The structure of this paper is discussed next. Firstly, the compressive and flexural strengths are reported. Needless to say, cement hydration activation should yield larger early age compressive strengths. The remaining part of the paper is intended to contribute to the understanding of the observed features. Thus, secondly, a calorimetric study is presented, where the pastes are studied up to seven days of hydration. Shorter induction periods, a larger slope in the acceleration stage, and higher heat flow peaks at early ages, ~5–15 h, are the three key signatures of very early age cement hydration acceleration. Thirdly, the thermal analysis results are reported for the hydrated pastes at 1, 7, and 28 days. Larger overall weight losses for the seeded samples indicate acceleration. Fourthly, the phase evolution with seeding is determined by Rietveld quantitative phase analysis in order to better understand the C-S-H seeding acceleration mechanism(s). Finally, the mercury intrusion porosimetry results are analysed. Here, the key descriptors are: (i) overall porosity and (ii) threshold pore entry size. Lower porosities, for a given maximum pressure at the same hydration time, and smaller threshold pore sizes signal enhanced hydration.

### 3.1. Mechanical Strengths

The mortar spread values, determined by the slump test, were 65 and 17 mm for PC-42.5 with *w*/*c* = 0.50 and 0.40, respectively. The corresponding spreads for BC-Buz were 73 and 20 mm and for BC-n.a., 112 and 38 mm. As expected, the spread values for the three mortars prepared with *w*/*c* = 0.40 are low. However, it was decided not to add superplasticizer (SP) in order to avoid the possible hydration delay due to the SP usage, which could partly mask the acceleration by the activator admixtures. This coupling effect will be researched in the future. With this information, Figure 1 displays the mortar mechanical strength data for the three studied types of mortar, including the seeded ones with 2 wt% of the XS130 and STE53 accelerators.

Concerning PC-42.5 with *w*/*c* = 0.50, in Figure 1a, the 1-day compressive strength increased 60% and 47% with the addition of XS130 and STE53, respectively. The corresponding increases at 7 days were 26% and 45%. These improvements are also mapped out by the flexural strength; see Figure 1b. Notably, the mechanical strengths at 28 days were maintained or even improved for STE53. On the other hand, for PC-42.5 with *w*/*c* = 0.40, in Figure 1c, the 1-day compressive also improved, but quantitatively less. The increase was 42% for both accelerators. XS130 did not improve the values at either 7 or 28 days. However, STE53 improved the compressive strengths at 7 and 28 days by 16% and 9%, respectively. The flexural strength values followed a very similar pattern; see Figure 1d.

For the activated belite cement, BC-Buz, with *w*/*c* = 0.50, the improvement of the compressive strength at the early ages is noticeable but smaller than those in PC-42.5. The values at 1 and 7 days increased by 29 and 19% for XS130 and by 41 and 21% for STE53. At 28 days there was not an improvement for STE53, but XS130 improved by 17%. Figure 1c shows the effect of the accelerators for BC-Buz, with *w*/*c* = 0.40, where the improvement in the mechanical strength values is much smaller. It is noted here that BC-Buz has not intentionally added limestone.

As expected, non-activated belite cement, BC-n.a., showed very low mechanical strengths at the early ages. For *w*/*c* = 0.50, the binders had 4 and 14 MPa at 1 and 7 days, respectively; see Figure 1a. A much larger value was obtained at 28 days, 46 MPa, because of the hydration of belite. The improvement with the use of the accelerators is minor but noticeable, 13%, at 28 days. In line with the results shown above for BC-Buz, the effect of the accelerators at lower *w*/*c* ratios, i.e., 0.40, is smaller; see Figure 1c.

Overall, the results are in line with the preliminary study carried out for just the XS130 accelerator for mortars with *w*/*c* = 0.50 [44]. The preliminary work has been extended here to a second accelerator and to a smaller and relevant *w*/*c* ratio.

### 3.2. Calorimetric Study

Figure 2a,b shows the heat flow and cumulative curves, respectively, for the PC pastes. Firstly, it can be seen that the behaviour of the two studied *w*/*c* ratio samples is quite similar, mainly for the pastes without accelerators. When comparing the seeded samples (XS130 and STE53) with the unseeded ones, the early age acceleration is evident, as expected [27]. This is deduced from: (i) the shorter ends of the induction periods (IP); (ii) the steeper slopes in the acceleration periods; (iii) the greater values of the maxima of the heat flow traces, located at about 8–10 h; and (iv) the earlier times for these maxima. This is usually explained by an enhanced alite hydration rate, but a partial contribution from a faster ettringite crystallization cannot be ruled out. In addition to the early age acceleration, the admixtures increased the height of the second peak, which is commonly assigned to an enhanced calcium aluminate dissolution, with the corresponding ettringite crystallization if enough sulfates are available [50]. It is also evident that accelerated cement hydration results in a narrowing of the heat flow peak at 8–12 h. We speculate that this behaviour is due to the enhanced aluminate dissolution, which is known to slow down alite hydration [51,52,53], i.e., yielding a steeper hydration rate decrease in the deceleration period. Pastes with the larger *w*/*c* ratio, i.e., 0.50, release more heat because the degree of hydrations of the different phases are slightly larger after one day; see Figure 2b. Finally, it is noticeable that the pastes seeded with STE53 released more heat at seven days than the ones seeded with XS130; see the inset of Figure 2a. This is in agreement with the larger compressive strength at that age, 49 and 43 MPa for the mortars containing STE53 and XS130, respectively.

Belite phase reactivity is slower than that of alite [4], and therefore, the heat flow and cumulative curves for BC-Buz show lower values; see Figure 2c,d. Interestingly, the heat flow curves for unseeded BC-Buz with the two studied *w*/*c* ratios are not very similar. The paste with *w*/*c* = 0.40 has a slightly larger IP and a slower heat development rate in the acceleration period. The hydration of both pastes is enhanced by the employed admixtures; see Figure 2c,d. At 7 days, and for a given *w*/*c* ratio, the paste seeded with STE53 released the largest heat; see the inset of Figure 2c.

Figure 2e,f displays the heat flow and cumulative curves for the BC-n.a. pastes. The unseeded pastes with the two studied *w*/*c* ratios show very similar heat release patterns. This behaviour is the same as that shown by the neat PC pastes, as can be seen in Figure 2a, but different from that displayed by the activated BC pastes; see Figure 2c. Again, the acceleration is evident mainly by a shortening of the IP and by a larger heat release at the early ages. At 1 day, the liberated heat increased from ~85 J/g to ~115 J/g. It is worth noting that the calcium aluminate dissolution was strongly affected by the admixtures. The STE53 seeding increased the aluminate peak, but it was still a shoulder of the main heat flow peak due to alite hydration. However, XS130 provoked a very sharp peak at about 17 h of hydration which was shown to be, for the *w*/*c* = 0.50 paste, an enhanced C_4_AF dissolution by in situ synchrotron X-ray powder diffraction [44]. This dissimilar behaviour is likely due to the different alkanolamines in the two employed admixtures. Coherently with the results for the other binders, enhanced aluminate dissolution provokes a slowdown of alite hydration which is reflected in the heat flow traces as narrower peaks with a steeper decrease in the deceleration periods. At 7 days, the heat released by the unseeded pastes, ~210 J/g, was increased by the admixtures by approximately 17%, to ~245 J/g.

### 3.3. Thermal Analysis Characterization

The thermal traces for the studies pastes, hydrated for 1, 7, and 28 days, are shown in Figure 3. Firstly, and as expected, for a given paste the total weight losses increase with the hydration age, reflecting the combined progression of the hydration reactions. Secondly, the overall cement hydration activation by the admixture is deduced from a larger weight loss at a given age, when seeded. For instance, PC-42.5-wc05-Ref has a total weight loss of 19.5% at 7 d. The corresponding losses for PC-42.5-wc05-XS130 and PC-42.5-wc05-STE53 are 21.7 and 21.8%, respectively. From this figure and for PC-42.5, the XS130 admixture is more effective, increasing the reactivity at an early age, i.e., 1 day of hydration; meanwhile, STE53 is more effective at longer hydration ages, i.e., 28 days. For instance, the weight losses of PC-42.5-wc05-XS130 and PC-42.5-wc05-STE53 at 28 days are 23.0 and 25.7%, respectively. However, the trend is reversed at 1 day, the values being 17.1 and 14.7%. For BCs, this pattern is not so clear and both admixtures seem to activate the cement hydration in a similar way at the studied hydration ages.

Thirdly, the portlandite contents, derived using the tangential method [49], are also given in Figure 3. Indeed, and for a given series, the CH content increased with the hydration time, revealing the progression of the alite and belite reactions. However, the portlandite contents did not increase; in fact, they slightly decreased at a given hydration time with C-S-H seeding. This behaviour has previously been reported for laboratory-prepared C-S-H seeds [54] and for X-Seed 130 additions [40,44]. This behaviour was justified in [40,54] by the assumption that the added C-S-H seeds in the pore solution can further stimulate the precipitation of C-S-H gel over the portlandite phase, resulting in a C-S-H gel with a larger Ca/Si ratio. An additional explanation, that could take place simultaneously, is the possible consumption of CH to yield additional AFm-type phases, both crystalline and/or amorphous. Fourthly, the minor weight loss within the ~600–800 °C range is due to the initial limestone content of the cements, and it indicated insignificant carbonation. Fifthly, the FW values are also given in Figure 3 for completeness; thus, they can be directly compared to the numbers already reported in the literature. FW follows an inverse trend to that of the total weight losses, but here, some effects, such as limestone (variable) contributions, are taken out. At a first approximation, a larger amount of FW is correlated with lower mechanical strength values as there is higher water porosity. Obviously, after cement activation the FW contents decreased; see the insets of Figure 3. However, different hydrated phases may consume different amounts of water and have different binding performances. This second order behaviour will be discussed in connection with the Rietveld quantitative phase analysis results.

### 3.4. Rietveld Quantitative Phase Analyses

LXRPD patterns for the 54 pastes and the three anhydrous cements were acquired with added quartz (the internal standard) in order to obtain the overall amorphous and crystalline-non-determined (ACn) contents [45,55]. Appendix A display, as examples, the Rietveld plots for the pastes with *w*/*c* = 0.40, after 28 days of hydration, for the three reference pastes (without admixtures) and the corresponding STE53-containing pastes. All the LXRPD raw data have been openly deposited, and hence, they could be inspected by any interested reader; see data availability subsection below.

The results of the Rietveld analyses are the six tables given in the Appendix A. These tables also contain the FW experimentally determined from the thermal investigation. Furthermore, some calculated contents are also given following the chemical reaction approach detailed in the Appendix A: (i) C-S-H gel content, (ii) additional amorphous phase(s), and (iii) remaining FW calculated according to the chemical reactions. These tables show the disappearance of the clinker phases as a function of time, and hence, the degrees of hydration (DoH’s) can be calculated. Furthermore, they also enable the detection of the (small) differences in the hydration pathways after adding the admixtures. The discussion of these results will be arranged in two subsections. Firstly, the DoH of the different clinker phases will be discussed. Secondly, the rate of appearance of the hydrated phases will be considered.

#### 3.4.1. Dissolution of the Clinker Phases

To have a better understanding of the role of the XS130 and STE53 admixtures in the cement hydration, Table 1 reports the DoH of the clinker phases. Several observations can be drawn from these data.

Alite/C_3_S. For PC-42.5 and the two studied *w*/*c* ratios, the DoH of the alite was approximately 55, 75, and 90% at 1, 7, and 28 days, respectively. This is fully in line with the known behaviour of PC hydration [56,57]. This study cannot conclusively prove that the reaction rate of crystalline alite in PC-42.5 is increased by C-S-H gel nanoparticle seeding at the studied hydration times, i.e., 1 day or later. For BC-Buz, with both *w*/*c* ratios, the alite reaction degree is approximately 65, 80, and 85% at 1, 7, and 28 days. Again, the alite DoH is not significantly accelerated by these admixtures. The slightly higher reaction degree at the very early ages, when compared to PC-42.5, is likely related to the lower amount of C_3_S, which results in higher water-to-alite mass ratio(s). For BC-n.a., the alite DoH was approximately 60, 85, and 90% at 1, 7, and 28 days, respectively. In agreement with the previous observations, C-S-H seeding does not seem to accelerate alite hydration. Finally, these results, for the studied PC and hydrating times, do not support the widely known picture that C-S-H seeding accelerates alite hydration. Faster alite hydration at the very early ages, i.e., earlier than 24 h, cannot be ruled out with the experimental data reported here.Belite/C_2_S. As can be seen in Table 1 for PC-42.5, the belite did not significantly react in the first week. This is again in line with the known features of ordinary PC hydration [56]. This is not the case for the BC-Buz pastes. For *w*/*c* = 0.50, the belite reacted about 5% at 7 days and, even more importantly, about 35% at 28 days. For *w*/*c* = 0.40, the belite hydrated about 10 and 40% at 7 and 28 days. This enhanced DoH of belite is due to the sulphur activation in the clinkering stage, which can be mapped out by a larger belite unit cell volume [58,59,60]. The scattering of the results did not allow a firm conclusion on any significant acceleration of belite phase hydration by XS130 or STE53. For the non-activated belite cement, the belite did not react at 7 days in any studied paste. Without admixtures, the DoH of belite for both *w*/*c* ratios was approximately 20% at 28 days. The admixtures seem to moderately accelerate belite hydration at 28 days, XS130 to 40% and STE53 to 30%. However, this isolated result needs additional confirmation.Tetracalcium alumino-ferrite/C_4_AF. For PC-42.5 and the two studied *w*/*c* ratios, the reaction degree of C_4_AF was approximately 10, 30, and 40% at 1, 7, and 28 days, respectively. The admixtures very significantly accelerated its hydration. For *w*/*c* = 0.50, both admixtures yielded a C_4_AF DoH of 20, 45, and 65%. For *w*/*c* = 0.40, the corresponding values were 20, 45, and 60%. Therefore, the acceleration of C_4_AF hydration by the employed admixtures is firmly established. On the other hand, C_4_AF reacts faster in BC-Buz. For the reference pastes, i.e., without admixtures, the DoH was 20, 40, and 75% (*w*/*c* = 0.50) and 15, 50, and 75% (*w*/*c* = 0.40) with hydrating time. In agreement with the results for PC-42.5, both admixtures accelerated C_4_AF hydration, the DoH being 55, 75, and 85% (*w*/*c* = 0.50) and 60, 75, and 85% (*w*/*c* = 0.40). Concerning the unseeded BC-n.a. pastes, the reaction degree of C_4_AF was approximately 15, 35, and 40% at 1, 7, and 28 days, respectively. Again, the admixtures strongly accelerated C_4_AF hydration as its degree of hydration increased to 25, 45, and 70% at the studied ages.Tricalcium aluminate/C_3_A. For PC-42.5 and both *w*/*c* ratios, the DoH of C_3_A was 7, 60, and 85% at 1, 7, and 28 days, respectively. The studied admixtures significantly accelerated C_3_A hydration. With some scattering in the results, the DoH of the studied pastes increased to 25, 65, and 90%. For BC-Buz, C_3_A had fully dissolved at 7 days. However, the hydration acceleration at 1 day was noteworthy. Finally, BC-n.a. did not contain C_3_A; hence, the role of the admixtures cannot be discussed.

#### 3.4.2. Precipitation of Hydrated Phases

To better understand the implications of the XS130 and STE53 admixtures in the hydrated phase development, the data reported in Appendix A have been plotted in Figure 4, Figure 5 and Figure 6. The data are always referenced to 100 g of paste. In addition to the dissolution (reaction) of the clinker phases, as has already been discussed, the formations of the hydrated phases are displayed.

Figure 4 (top and bottom) shows the phase evolutions of the pastes for PC-42.5-wc05 and PC-42.5-wc04, respectively. It is worth noting that the Minors-C component groups all minor phases in the cement, i.e., calcium sulfate(s), limestone, etc. As expected, the hydration pathway of PC was affected to a minor extent by the addition of 2 wt% of the admixtures, XS130 or STE53. The largest difference, concerning the crystalline phases, was the promotion of the crystallization of the AFm-type phases by the admixtures. For both *w*/*c* ratios and admixtures, larger amounts of hemi (Hc) and mono (Mc) carbonate phases were quantified, mainly at 28 days of hydration. Concerning ettringite, the admixtures slightly promoted its crystallization at 1 day, but the ettringite contents at 28 days were smaller in the admixture containing pastes. This is compatible with the observed slightly lower amount of portlandite in the admixture containing phases (measured by thermal analysis and RQPA). A possible explanation could be the reaction of a fraction of Ca(OH)_2_ with ettringite, in the presence of carbonate anions from the limestone, to yield Hc and Mc phases. This could also imply that the optimum sulphate content of the binders, when using these admixtures, could be slightly higher. However, the optimum sulphate content related to the C-S-H admixture dosage requires further investigation. Finally, the ACn phase evolution deserves attention. This component groups all amorphous phases, but the contributions from free water and C-S-H were taken out and separately represented. The typical phases grouped within ACn are iron siliceous hydrogarnet (from C_4_AF hydration) and amorphous AFm phases. Figure 4 shows that the amounts of amorphous phases, ACn, are significantly larger in the admixture containing pastes. This is fully in line with a higher C_4_AF DoH, discussed in the previous subsection, which should yield a larger amount of amorphous iron siliceous hydrogarnet.

Figure 5 shows the phase developments for BC-Buz with *w*/*c* = 0.50 and 0.40, without and with the XS130 and STE53 admixtures. In agreement with the observations discussed above, the admixtures containing pastes have larger amounts of the AFm-type phases, which are Hc and Mc. The ettringite content in the activated pastes is only larger at 1 day of hydration. Finally, and importantly, the amounts of ACn are larger in the admixture containing pastes than in the neat ones. This is due, at least in part, to a larger C_4_AF reactivity when the admixtures are employed.

Figure 6 shows the phase development for BC-n.a. with *w*/*c* = 0.50 and 0.40 with the used admixtures. It should be noted that this cement does not have added limestone; hence, the very small amount of measured Hc, always smaller than 1 wt%, arises from the carbonation of the pastes. Concerning the development of the hydrated phases, the most noteworthy feature is again a larger amount of ACn for the admixtures containing pastes.

It is acknowledged that the ACn content of the employed PC-42.5 cement, i.e., ~13 wt%, is high; see Appendix A. It is also noted that this PC does not have any crystalline alkaline sulphate content. Moreover, this non-accounted content also has a contribution from the limestone addition to the cement, which usually contains crystalline phases in minor amounts that are diluted and therefore not measured in the analyses. The determined ACn value is larger than the reported ones, commonly ranging from 4–8 wt% [55,61,62,63]. The implications of a relatively large ACn content with respect to C-S-H seeding are not known at this moment. Research on the C-S-H seeding of PCs with variable phase assemblages, including different ACn and sulfate contents, is planned, and it will be reported elsewhere.

### 3.5. Mercury Intrusion Porosimetry Study

Mercury only percolates through connected porosity at a given pressure, which leads to an underestimation of the biggest (unconnected) pores. This effect is known as the ink-bottle effect [49], and it is one of the main drawbacks of this technique for microstructural characterization. However, it is simple to use and when applied to similar series under exactly the same experimental conditions and data treatment assumptions, it can yield valuable information. For simplicity, the term pore size is employed here when referring to MIP-determined pore entry sizes. Next, the analysis of the total porosity is determined at the highest available pressure, and the threshold pore entry radius is carried out [49]. Figure 7 displays the cumulative porosities for all studied pastes at 1, 7, and 28 days of hydration. In addition to the decrease in porosity with the hydration time and smaller *w*/*c* ratio, the consequence of the admixtures in the microstructure development can be mapped out.

For PC-42.5-wc05 at 1 day, in Figure 7a, the admixtures slightly reduced the total porosities of the pastes. Moreover, XS130 reduced the threshold pore size more than STE53. This behaviour was retained at 7 days, but at 28 days, the situation was reversed. For PC-42.5-wc04, in Figure 7b, the total porosities are obviously smaller. However, the role of the admixtures was maintained and XS130 was more effective in decreasing the threshold pore radii at the early ages. In line with the results shown previously for *w*/*c* = 0.50, the paste containing STE53 showed the smallest total porosity at 28 days.

For BC-Buz-wc05, in Figure 7c, the admixtures again reduced the total porosities of the pastes. At 1 day, the XS130 and STE53 additions resulted in similar threshold pore radii, but STE53 seemed to yield slightly lower total porosity. At 7 days, XS130 gave the smallest threshold pore radius and total porosity. At 28 days, both admixtures reduced the total porosities with very similar traces. For BC-Buz-wc04, in Figure 7d, similar conclusions can be drawn. Both admixtures reduced the total porosities and the threshold pore radii. For this low water–cement ratio, XS130 seems to be a bit more effective in decreasing the total porosities, and the threshold pore values are the same for both admixtures within the variability of the measurements.

For the BC-n.a.-wc05 pastes, in Figure 7e, the total porosities at early ages are the largest, which is a consequence of their lower hydration rate; see above. At 1 day, the role of the admixtures was minor, but at 7 and 28 days XS130 again seemed to be more effective in decreasing the total porosities. For the BC-n.a.-wc04 pastes, in Figure 7f, it can be concluded that XS130 is slightly more effective than STE53 for reducing the threshold pore radii. On the other hand, the decrease in total porosities by both admixtures is similar.

## 4. Conclusions

The employed admixtures, X-Seed 130 and X-Seed STE53, enhanced the cement hydration rate of Portland and belite cements, which is evidenced in the calorimetric curves by shorter induction periods, steeper slopes in the acceleration period, and larger heat flow values of the first peak of hydration. Cement hydration acceleration is also evidenced by the thermal analyses, because for the same hydration age the pastes containing the admixtures have smaller free water contents and therefore larger combined water fractions.

For both studied *w*/*c* ratios, 0.50 and 0.40, the Rietveld quantitative phase analyses at 1, 7, and 28 days indicated that the alite and belite degree of hydrations, derived from the crystalline content variation, were not significantly enhanced by C-S-H seeding. Conversely, the tetracalcium alumino-ferrate and tricalcium aluminate hydrations were accelerated by the admixtures at the studied hydration times. This behaviour was justified because of the synergistic effect of the C-S-H seeding with alkanolamines. C-S-H nanoparticles provide extra surfaces for aluminium species adsorption, which accelerates the dissolution of these phases. Furthermore, these admixtures contain alkanolamines which can contribute to the increased C_4_AF reactivity as alkanolamines are known to coordinate different cations, including iron. In line with this behaviour, an enlarged amount of amorphous phases, which includes amorphous iron siliceous hydrogarnet, was measured in the admixtures containing pastes. The behaviour of the non-diffracting/amorphous components of the cements with C-S-H seeding deserves further investigation.

From the mechanical strength perspective for PCs, the Master X-Seed 130 admixture develops a larger increase at the early ages; meanwhile Master X-Seed STE53 helps to develop larger strengths at 28 days. This pattern is observed for both of the studied *w*/*c* ratios, 0.50 and 0.40. Concerning the roles of these admixtures for accelerating the mechanical strength development in belite cements, the situation is more scattered. More research is needed to establish the potential of this type of accelerator in belite cements.

## Figures and Tables

**Figure 1 materials-15-03553-f001:**
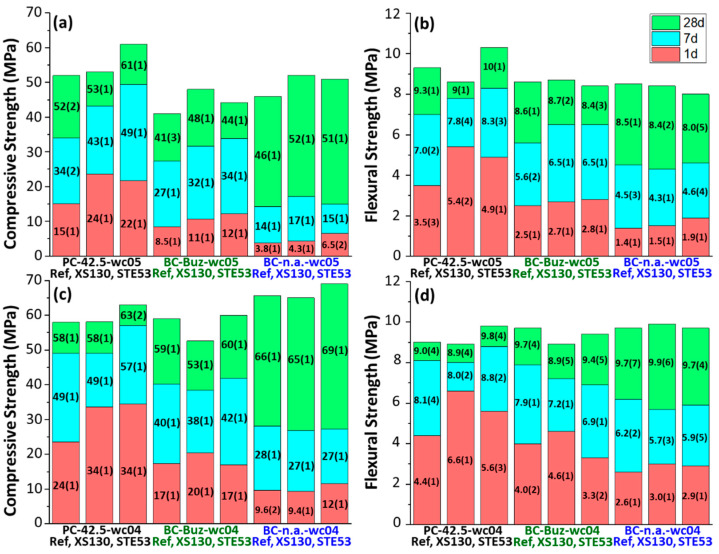
Mechanical strength results for the eighteen studied mortars at 1, 7, and 28 days of hydration. Compressive strengths (left panels: (**a**,**c**)). Flexural strengths (right panels: (**b**,**d**)).

**Figure 2 materials-15-03553-f002:**
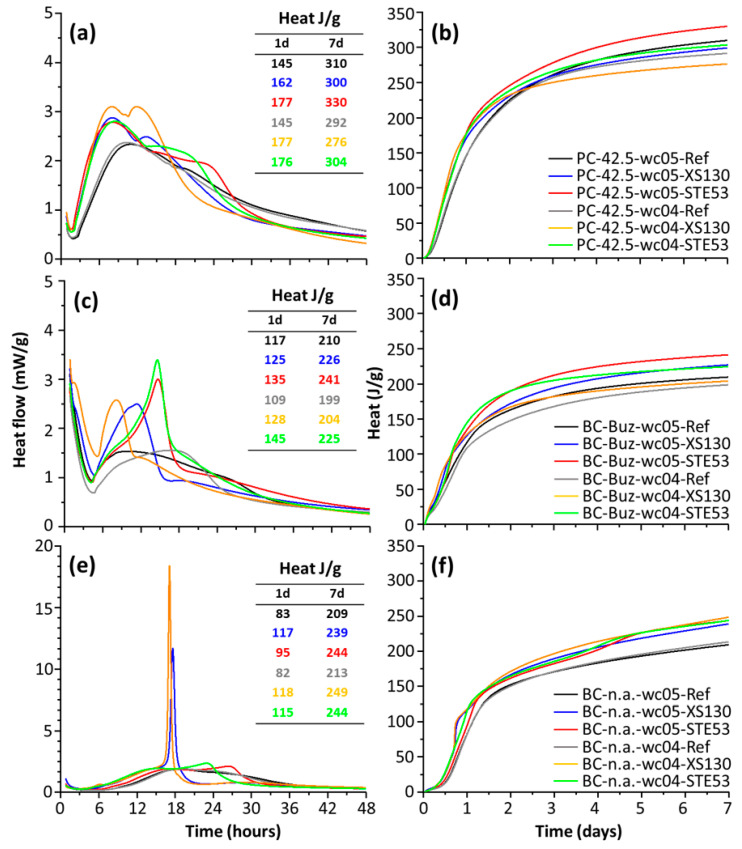
Calorimetries for the eighteen studied pastes, showing the outcomes of the employed accelerator admixtures. The heat flow curves, left panels, are referenced to 100 g of cement and only the first 48 h is shown for clarity. The total heat released, right panels, during the first seven days of hydration. (**a**,**b**) PC pastes, (**c**,**d**) BC-Buz pastes, (**e**,**f**) Non-activated BC pastes. The insets report the heats developed at 1 and 7 days by the studied pastes.

**Figure 3 materials-15-03553-f003:**
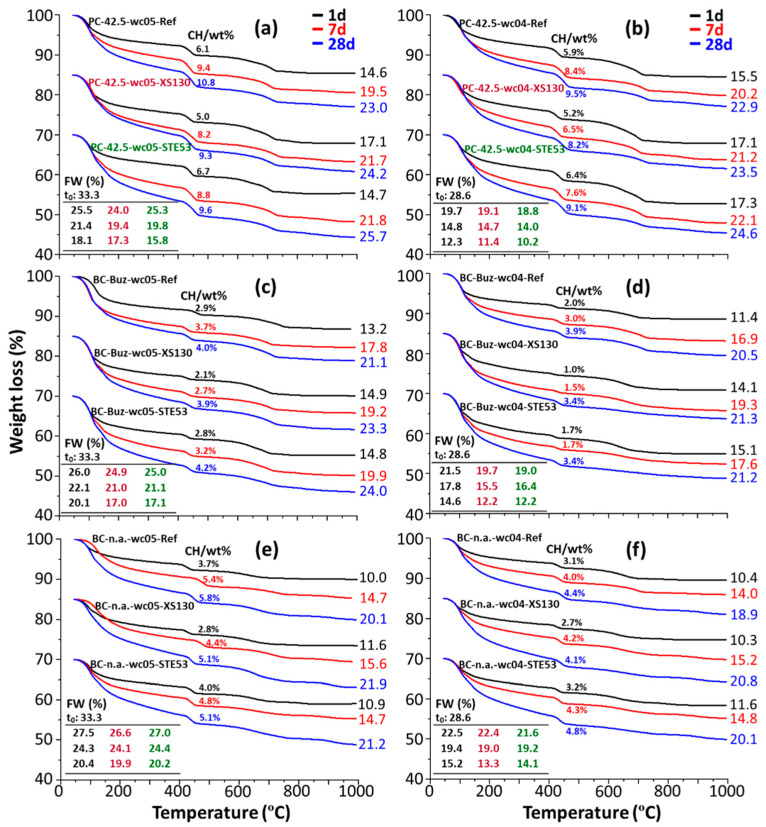
The fifty-four thermal analysis traces for the pastes after arresting the hydration and holding 30 min at 40 °C. The curves for the series have been vertically displaced for clarity. (**a**) PC-42.5 *w*/*c* = 0.50; (**b**) PC-42.5 *w*/*c* = 0.40; (**c**) BC-Buz *w*/*c* = 0.50; (**d**) BC-Buz *w*/*c* = 0.40; (**e**) BC-n.a. *w*/*c* = 0.50; (**f**) BC-n.a. *w*/*c* = 0.40. The Portlandite contents (wt%) given refer to the neat pastes, i.e., containing the free water, for direct comparison with the RQPA results. The total weight losses (wt%) are also displayed to the right of the traces. The insets give the free water contents, determined as described in the experimental section.

**Figure 4 materials-15-03553-f004:**
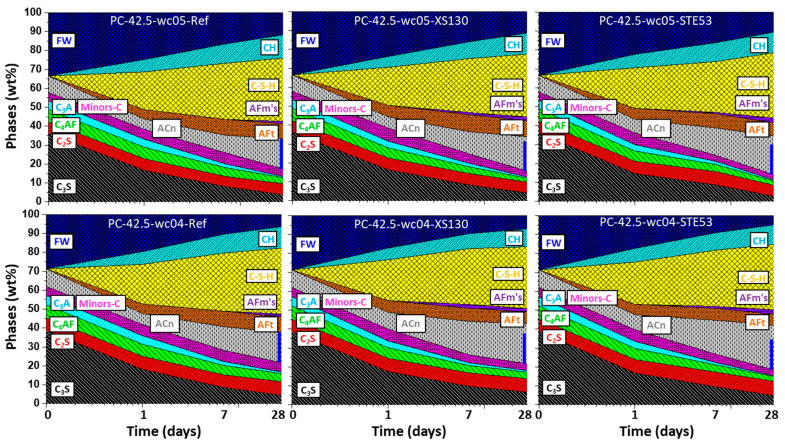
Phase development for PC-42.5 pastes based on the RQPA results: (**top**) *w*/*c* = 0.50, (**bottom**) *w*/*c* = 0.40. The total amorphous content determined by the internal standard methodology has been divided into three components: calculated free water, C-S-H content, and ACn, as described in the Appendix A; these amounts are obtained taking into account the chemical hydration reactions. ACn accounts for any other amorphous phases, i.e., iron siliceous hydrogarnet, AFm-type phases, and the amorphous content within the unreacted clinker phases. The blue vertical lines highlight the amount of ACn at 28 days in the two pastes without admixtures. The seeded pastes have significantly higher amounts of amorphous phases; see the blue lines. Minors-C stands for all minor phases in the pristine cement.

**Figure 5 materials-15-03553-f005:**
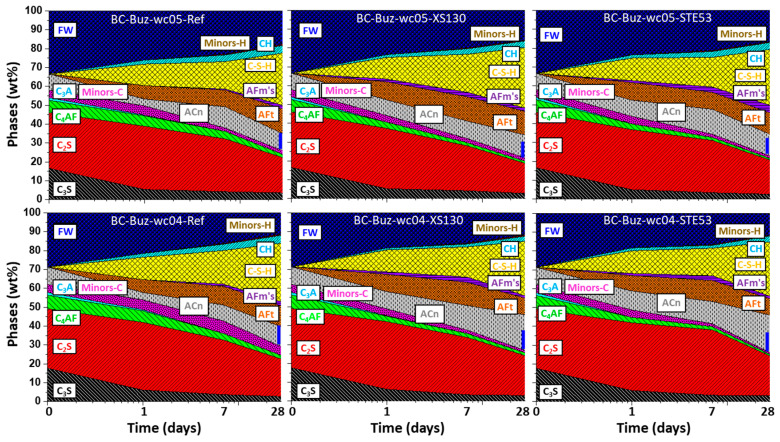
Phase development for BC-Buz (**top**) *w*/*c* = 0.50, (**bottom**) *w*/*c* = 0.40, with all details as in Figure 4. Minors-H stands for any minor hydrated phase not accounted for in the explicitly given ones.

**Figure 6 materials-15-03553-f006:**
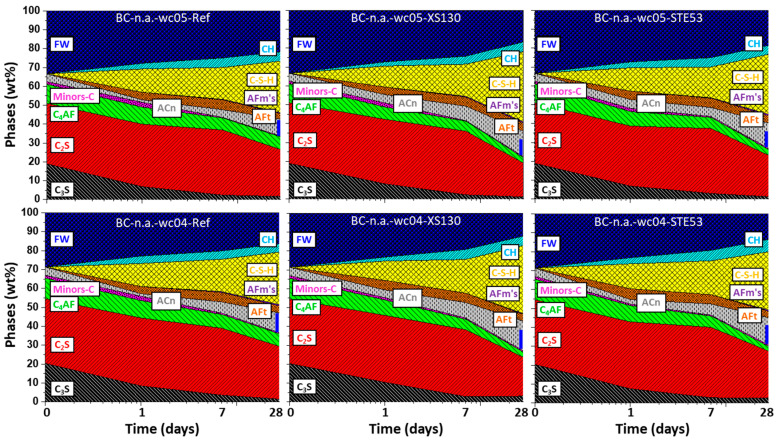
Phase development for BC-n.a. (**top**) *w*/*c* = 0.50, (**bottom**) *w*/*c* = 0.40, with all details as in Figure 5.

**Figure 7 materials-15-03553-f007:**
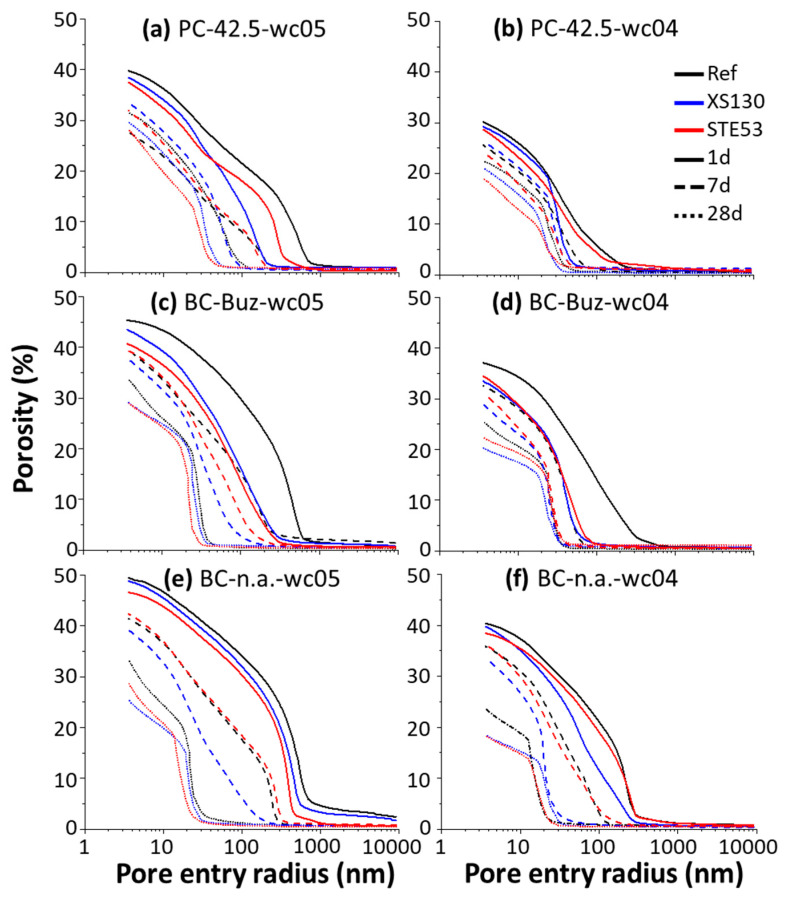
Mercury intrusion porosimetry cumulative porosity curves for the eighteen studied pastes at 1, 7, and 28 days of hydration. Left panels (**a**,**c**,**e**) display data for the *w*/*c* = 0.50 pastes. Right panels (**b**,**d**,**f**) show data for the *w*/*c* = 0.40 pastes.

**Table 1 materials-15-03553-t001:** Degree of hydration of the clinker phases for all studied pastes.

Sample	DoH C_3_S (%)			DoH C_2_S (%)			DoH C_4_AF (%)			DoH C_3_A (%)		
	1 d	7 d	28 d	1 d	7 d	28 d	1 d	7 d	28 d	1 d	7 d	28 d
PC-42.5-wc05-Ref	54	78	89	2	6	6	8	30	50	7	56	89
PC-42.5-wc05-XS130	55	77	88	-	3	8	18	45	68	22	60	87
PC-42.5-wc05-STE53	59	76	93	-	-	6	18	42	65	27	64	89
PC-42.5-wc04-Ref	53	78	88	-	-	-	9	29	39	6	58	83
PC-42.5-wc04-XS130	55	75	83	-	-	-	20	44	50	25	71	83
PC-42.5-wc04-STE53	57	75	87	-	-	-	16	46	63	23	75	94
BC-Buz-wc05-Ref	67	75	78	-	5	36	18	40	75	69	100	100
BC-Buz-wc05-XS130	68	75	83	-	19	46	51	75	84	100	100	100
BC-Buz-wc05-STE53	70	80	84	-	6	38	54	76	87	100	100	100
BC-Buz-wc04-Ref	66	79	85	-	8	37	15	51	74	57	100	100
BC-Buz-wc04-XS130	64	81	83	-	3	33	60	71	79	79	100	100
BC-Buz-wc04-STE53	67	81	82	-	-	33	60	75	89	100	100	100
BC-n.a.-wc05-Ref	64	87	91	-	-	22	11	37	34	-	-	-
BC-n.a.-wc05-XS130	57	88	94	-	-	44	36	48	71	-	-	-
BC-n.a.-wc05-STE53	64	84	93	-	-	32	24	42	69	-	-	-
BC-n.a.-wc04-Ref	59	82	93	-	-	18	17	33	39	-	-	-
BC-n.a.-wc04-XS130	49	86	85	-	-	40	25	49	73	-	-	-
BC-n.a.-wc04-STE53	63	86	88	-	-	27	21	45	76	-	-	-

## Data Availability

Raw data derived from the following techniques, isothermal calorimetry, thermal analysis, laboratory X-ray powder diffraction, and mercury intrusion porosimetry, may be openly accessed on Zenodo at https://doi.org/10.5281/zenodo.6335735 (accessed on 15 May 2022), and used under the Creative Commons Attribution license. Full details of the different datasets are given in the Appendix A.

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
