# Peer review of "Portland and Belite Cement Hydration Acceleration by C-S-H Seeds with Variable w/c Ratios"

_materials, 2022, doi:10.3390/ma15103553_

Round 1

Reviewer 1 Report

This study is interesting.

  1. Wrong written,such as Initial case problem “Rietveld quantitative phase” in Line21ï¼›

2. Why have literatures bee ncited in Conclusions“This behaviour is justified because the 540 synergistic effect of C-S-H seeding with alkanolamines [19,39].”?

3. What is S.I. in Line200?

4. Wrong grammar,"SCMs being nowadays the procedure with lowest performance and economic impact" in line 40

Author Response

See attached letter

Reviewer 2 Report

This is a very interesting study dealing with Portland and Belite cement hydration acceleration by C-S-H seeds with variable w/c ratios. This manuscript presents a specific, easily identifiable advance in knowledge. It is applicable and useful to the profession. The title and abstract accurately describe the contents. In the manuscript, a comprehensive literature review was conducted, and all references are pertinent and complete. The main aim of the paper is clearly stated, and the methodology is sufficiently well explained that someone else knowledgeable about the field could repeat the study although subject is quite detailed and complicated; language used in article is fluent. Classifications in tables and figures clearly represent experimental studies conducted before. Each figure and table are necessary to the understanding of the conclusions. The results are soundly interpreted and related to existing knowledge on the topic. The conclusions are sound and justified. They follow logically from data presented. All elements of the manuscript relate logically to the study's statement of purpose. As a conclusion, the current version of the manuscript can be suitable for publication in Materials journal with some minor revisions.

Author Response

See attached letter

Reviewer 3 Report

This study aims to study the effects of C-S-H seeding in one Portland cement and two belite cements at two widely-studied water-cement ratios, 0.50 and 0.40.  alite and belite phase hydrations, from the crystalline phase content evolutions, are not significantly accelerated by C-S-H seedings at the studied ages of 1 and 28 d for these cements.

The article is well research and contains novel idea that adds some information to the body of knowledge. Likewise, the paper complies with the writing standard of the Journal and all tests were done according to the normal standard of tests. Based on these aforementioned, I recommend that the research paper can be accepted for publication after the minor revisions is corrected according to suggestions.

  1. The information related to the mineral composition of raw materials is very important. Please supplement it, such as particle size analysis, density, specific surface area and so on.
  2. The author expounds the mineral composition of raw materials in 129-130. The relevant composition can be seen in tables S1, S2 and S3, but the relevant tables are not found.
  3. Are the two CSH nano materials or ordinary materials?
  4. Where is table s4-s9?
  5. In the article, the author talked about using CSH as seed crystal to promote hydration. Generally, the most seed crystal material is nano material, and nano material needs to consider its dispersion. Does the author consider it.

Author Response

See attached letter

Reviewer 4 Report

This study investigates Two commercially available C-S-H based admixtures. A multi-technique approach is adopted by employing: calorimetry, thermal analysis, Rietveld quantitative phase analysis, mercury intrusion porosimetry and mechanical strength determination. The evolution of the amorphous contents was determined by adding an internal standard before recording the powder patterns. Results showed that the admixtures yielded larger amounts of amorphous components in the pastes at later hydration ages. The abstract is well-written as the abstract required to have a short introduction, problem statement, significant finding, and conclude the abstract with your outcome or novelty. The conclusion has been written well and tightened to reveal the overall finding. The paper is fruitful and of interest to future readership and includes information on an interesting topic, it is both well written and well organized. Although the testing methods and compared results attained in the present study show the importance of the paper, the authors should address the following comments:

  1. Throughout the text, there are some typos that must be eliminated.
  2. There should be a space between number and unit. Please correct these errors in the paper.
  3. 30 % should be 30% in Line# 137.
  4. EN196-1 should be EN 196-1 in Line# 157 and so on.
  5. Please change Figure 2 with a better quality one.

Author Response

See attached letter
